# Defective apical extrusion signaling contributes to aggressive tumor hallmarks

**Yapeng Gu[1]\*, Jill Shea[2], Gloria Slattum[1], Matthew A Firpo[2], Margaret Alexander[1], Sean J Mulvihill[2], Vita M Golubovskaya[3], Jody Rosenblatt[1]\***

[1]Huntsman Cancer Institute, University of Utah, Salt Lake City, United States; [2]Department of Surgery, University of Utah, Salt Lake City, United States; [3]Department of Surgical Oncology, Roswell Park Cancer Institute, Buffalo, United States

**Abstract** When epithelia become too crowded, some cells are extruded that later die. To extrude, a cell produces the lipid, Sphingosine 1-Phosphate (S1P), which activates $S1P_2$ receptors in neighboring cells that seamlessly squeeze the cell out of the epithelium. Here, we find that extrusion defects can contribute to carcinogenesis and tumor progression. Tumors or epithelia lacking $S1P_2$ cannot extrude cells apically and instead form apoptotic-resistant masses, possess poor barrier function, and shift extrusion basally beneath the epithelium, providing a potential mechanism for cell invasion. Exogenous $S1P_2$ expression is sufficient to rescue apical extrusion, cell death, and reduce orthotopic pancreatic tumors and their metastases. Focal Adhesion Kinase (FAK) inhibitor can bypass extrusion defects and could, therefore, target pancreatic, lung, and colon tumors that lack $S1P_2$ without affecting wild-type tissue.

\*For correspondence: yapeng. gu@hci.utah.edu (YG); jody. rosenblatt@hci.utah.edu (JR)

**Competing interests:** The authors declare that no competing interests exist.

**Reviewing editor**: Ewa Paluch, University College London, United Kingdom

## Introduction

Epithelial cells must act collectively to provide a protective barrier for the organs they encase even though they continuously turn over through cell death and division. The link between cell division and death is critical: if the relative death rate is too high, barrier function diseases may result whereas if division outpaces cell death, epithelia could become neoplastic. We previously identified a process critical for promoting cell death when cells within epithelia become overcrowded termed epithelial extrusion (*Rosenblatt et al., 2001*; *Eisenhoffer et al., 2012*). The stretch-activated channel Piezo-1 senses cell crowding and enables some cells to produce the bioactive sphingolipid, Sphingosine 1-phosphate (S1P), which binds G-protein coupled receptors ($S1P_2$) in neighboring cells to activate Rho-mediated assembly and contraction of an intercellular actomyosin ring (*Gu et al., 2011*). This contraction squeezes live cells apically out of the epithelial sheet while simultaneously closing the gap that might have resulted from the cell's exit, thus preserving epithelial barrier function. Because live extruded cells become stripped from the underlying matrix and its associated survival signaling, they later die by anoikis (*Frisch and Francis, 1994*).

Advanced tumors typically have increased survival signaling that overrides anoikis, suggesting that cells could survive following extrusion. In this case, the direction a cell extrudes can impact its later fate. Typically, epithelia extrude cells apically into the lumen of the tissue (*Slattum et al., 2009*), which would act to essentially eliminate tumor cells with upregulated survival signaling. However, some cells are extruded basally into the tissue encased by the epithelium (*Slattum et al., 2009*). If basally extruded cells survive following extrusion, they might be able to invade into the underlying tissue (*Slattum and Rosenblatt, 2014*). Interestingly, we have found that oncogenic mutations in either adenomatous polyposis coli or K-Ras misregulate apical extrusion and drive extrusion basally (*Marshall et al., 2011*; *Slattum et al., 2014*).

**eLife digest** Epithelial cells cover the surface of our bodies, line our lungs, stomach and intestines and serve as a protective layer around other organs. If too many epithelial cells die and are not replaced, this protective layer may erode and lead to organ damage. However, if too many new cells grow, tumors can form.

One process that helps to maintain the right number of epithelial cells is called extrusion. When too many epithelial cells are present, the resulting overcrowding triggers this process to squeeze excess cells out of the layer and away from the organ. Usually, these cells quickly die. However, if the pathway that regulates this process—which involves a receptor protein called $S1P_2$—is disturbed, the cells may instead be pushed into the space between the epithelial layer and the organ. When this happens, the cells are more likely to survive and may then form a tumor that invades the organ.

Gu et al. interfered with extrusion by reducing the levels of the $S1P_2$ receptor in layers of human epithelial cells grown in the laboratory. Fewer epithelial cells were squeezed out of these cell layers, making the layers up to three times as thick in places. Moreover, mutant zebrafish lacking the $S1P_2$ receptor also accumulated epithelial masses throughout their bodies.

Gu et al. found that disrupting the extrusion process made the cells resistant to chemotherapy, and that certain hard-to-treat human pancreatic, lung, and colon cancers had lower levels of the $S1P_2$ receptors. Boosting the activity of $S1P_2$ receptors helped to restore normal extrusion and reduced the size of pancreatic tumors in mice.

Gu et al. then focused on an enzyme called Focal Adhesion Kinase that helps cells to survive. Treating zebrafish with a drug to block the activity of this enzyme left normal fish unharmed. However, in mutant fish with malfunctioning extrusion pathways, the drug rescued the number of cells that died, reduced the size and number of masses, and cured their leaky skin barrier. If further studies confirm the results, the drug may offer a new, less toxic, treatment for certain cancers that do not respond to currently available treatments.

Here, we examined the long-term effects of disrupting the $S1P$-$S1P_2$ epithelia extrusion-signaling pathway. We found that inhibition of $S1P_2$ leads to large epithelial masses in both zebrafish epidermis and cultured epithelia and increased rates of basal extrusion. Moreover, disrupting extrusion by a variety of methods leads to chemotherapy resistance. Inhibition of Focal Adhesion Kinase (FAK), a key survival signal generated from cell-matrix adhesion, selectively promotes apoptosis in cells where extrusion is defective and eliminates epidermal cell masses formed in zebrafish $S1P_2$ mutants without affecting epithelial morphology and function. Pancreatic Ductal Adenocarcinomas (PDACs) have little to no $S1P_2$, which could explain why these tumors are typically more invasive and chemo-resistant. HPAF II human pancreatic cancer cells cannot extrude apically and instead extrude basally, survive, and proliferate following extrusion. Ectopically expressing $S1P_2$ in HPAF II cells rescues apical extrusion and apoptosis and reduces orthotopic mouse tumors and their metastases. Together, our results suggest that defective extrusion may be a new mechanism for how PDACs and other carcinomas lacking $S1P_2$ initiate and invade. Furthermore, FAK inhibitors, which are currently in clinical trials for other tumors, may provide an effective therapeutic opportunity to treat pancreatic cancer without destroying nearby normal tissue.

## Results

### Disruption of extrusion signaling reduces epithelial apoptosis rates

To test if the $S1P$-$S1P_2$-Rho signaling pathway that controls extrusion (*Gu et al., 2011*) was critical for preventing neoplastic growth over time, we knocked down $S1P_2$ in Human Bronchial Epithelial (HBE) cells (*Figure 1A*) and grew them for up to 3 weeks after they formed an intact monolayer. $S1P_2$-depleted HBE epithelia, which are extrusion-deficient (*Gu et al., 2011*), accumulated into masses over three layers thick whereas control-knockdown monolayers retained a single layer (*Figure 1B*). Because cell extrusion typically promotes epithelial cell death (*Eisenhoffer et al., 2012*; *Marinari et al., 2012*) and because $S1P_2$ depletion did not affect the proliferation rate in a yellow tetrazolium MTT (3-(4, 5-dimethylthiazolyl-2)-2, 5-diphenyltetrazolium bromide) assay (*Figure 1C*), masses were likely to

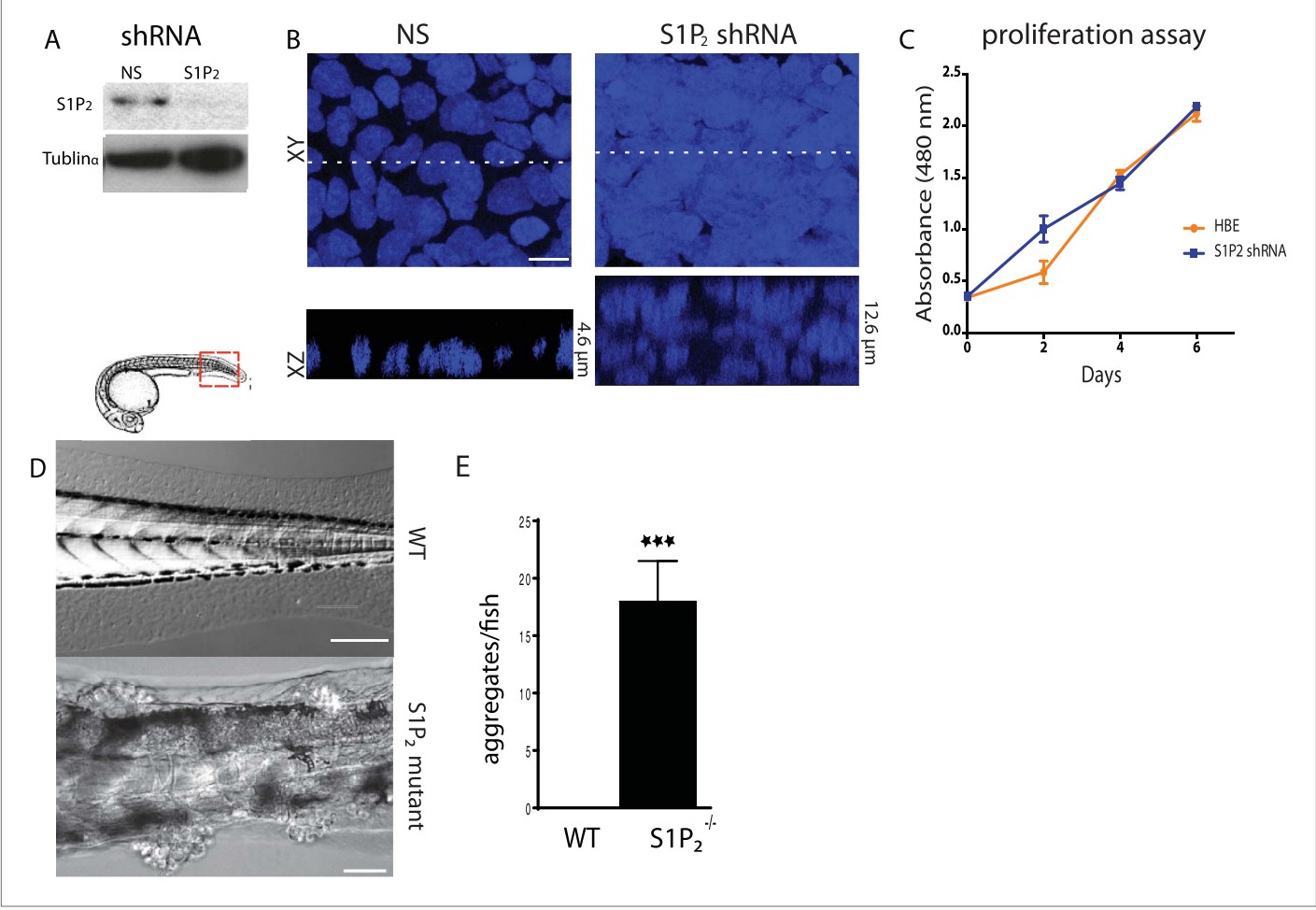

**Figure 1**. Loss of S1P$_2$ and extrusion leads to accumulation of epithelial cell masses. (**A**) S1P$_2$ immunoblot of HBE cells expressing control (left) or S1P$_2$-specific shRNA (right) with α-tubulin as loading control. (**B**) Representative images of HBE cells (DNA only) expressing control (left) or S1P$_2$-specific shRNA (right) grown for 3 weeks. Scale bar, 10 µm. (**C**) Proliferation assay indicates that S1P$_2$-knockdown cells proliferate at the same rate as wild type controls cells. (**D**) Representative DIC micrographs of 5-dpf WT (top) and Mil (S1P$_2$ mutant) (bottom) zebrafish larvae, where cartoon shows region where fish was imaged. Scale bars, 100 µm where red box indicates region imaged. (**E**) Quantification of epidermal clumps of 22 zebrafish larvae.

arise due to reduced apoptosis. Additionally, zebrafish larvae carrying a loss-of-function mutation in S1P$_2$ (Miles apart [*Mil*]) cannot extrude apoptotic epidermal cells (*Gu et al., 2011*), and similarly accumulated numerous epidermal cell aggregates throughout the body (18 ± 3.5 aggregates/fish, n = 22) by only 5 days post fertilization (dpf) (*Figure 1D,E*). By contrast, masses were undetectable in heterozygote *Mil* or WT siblings of the same age (*Figure 1D,E*).

We next wondered if extrusion-deficient cells were also more resistant to cell death in response to apoptotic stimuli. While extrusion promotes apoptosis during normal homeostasis by extruding live cells that later die from loss of contact to matrix-derived survival signaling (*Eisenhoffer et al., 2012*), treating epithelia with apoptotic stimuli causes cells to simultaneously die and extrude (*Rosenblatt et al., 2001*; *Andrade and Rosenblatt, 2011*). Because extrusion normally drives cell death, could it also help promote apoptosis in response to apoptotic stimuli by eliminating competing survival signaling associated with the underlying matrix? We find that disrupting extrusion signaling also disrupted apoptosis in response to a variety of apoptotic stimuli. HBE monolayers lacking S1P$_2$ (*Figure 2A*) or treated with a selective S1P$_2$ receptor antagonist, JTE-013 (*Figure 2B*) had greatly reduced rates of apoptosis in response to a strong apoptotic stimulus, UV-C, compared to controls. Madin–Darby Canine Kidney (MDCK) monolayers treated with S1P$_2$ antagonist were similarly resistant to several common chemotherapy drugs that cause apoptosis (*Figure 2B,C*).

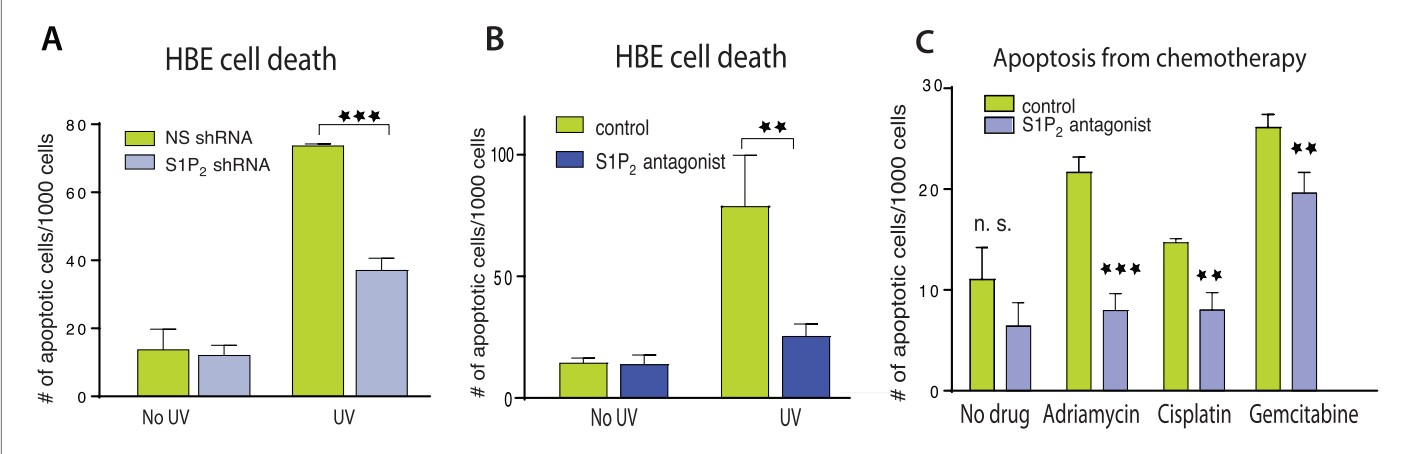

**Figure 2**. Disruption of $S1P_2$-extrusion signaling reduces apoptotic response. (**A**) Quantification of UV-induced apoptotic cells in HBE monolayers expressing control or $S1P_2$-specific shRNA. (**B**) Quantification of UV-induced apoptosis of HBE monolayers in the presence or absence of the $S1P_2$ antagonist JTE-013. (**C**) Quantification of indicated chemotherapy-induced apoptotic MDCK cells in the presence or absence of JTE-013, where all error bars are STD (**p < 0.01, and ***p < 0.001).

The reduced cell death rates in epithelia lacking $S1P_2$ were due to disruption of extrusion rather than altered S1P signaling, since other inhibitors of extrusion, Rho kinase inhibitor (Y-27632), myosin II inhibitor (Blebbistatin), or Rac inhibitor (EHT1864) all decreased cell death rates to the extent that they inhibit extrusion (**Figure 3A**). In each case, the ratio of cell death to extrusion inhibition is ~1:1 (**Figure 3C**). Inhibition of apoptosis was not due to increasing levels of S1P, which can act as a pro-survival signal, as S1P levels in apoptotic cells varied independently of extrusion inhibition (**Figure 3B**). Since freshly plated single MDCK cells are resistant to apoptotic stimuli, we tested if these same compounds reduced apoptosis in similarly aged single MDCKs by treating with EGTA to disrupt cadherin-dependent cell–cell contacts. Inhibitors that blocked apoptosis by blocking extrusion in an intact monolayer do not impact the apoptosis rates of single cells that are incapable of extrusion (**Figure 3D**). Similarly, UV-induced apoptosis was unaltered in single HBE cells lacking $S1P_2$ when HBE monolayers where treated with EGTA (**Figure 3D**). Additionally, inhibiting $S1P_2$ with JTE-013 in a cell line that cannot extrude but expresses this receptor (**Clair et al., 2003**; **Pham et al., 2013**), NIH 3T3 fibroblasts, does not affect the cell death rate in response to UV-C (**Figure 3E**). These data together suggest that increased cell survival is linked with the inability to extrude rather than to any intrinsic block of the apoptosis pathway.

## Pancreatic cancer cells lack the $S1P_2$ receptor and extrude basally rather than apically

Since disruption of $S1P_2$ in epithelia results in reduced apoptosis and cellular masses both in vitro and in vivo, we wondered if this receptor might be deficient in carcinomas. Our analysis of published tumor microarray data found $S1P_2$ mRNA to be significantly reduced in PDAC (**Buchholz et al., 2005**; **Segara et al., 2005**; **Badea et al., 2008**), and some lung and colon tumors (**Bhattacharjee et al., 2001**), compared to their corresponding normal tissues. To investigate if cancer cells lacking $S1P_2$ also have extrusion and apoptosis defects, we analyzed a pancreatic adenocarcinoma cell line, HPAF II, that has reduced $S1P_2$ levels (**Figure 4A**) and forms epithelial monolayers necessary for assaying extrusion. We used MDCK and HBE cells as controls, which are well characterized in several extrusion studies (**Rosenblatt et al., 2001**; **Slattum et al., 2009**; **Gu et al., 2011**), as the only immortalized normal pancreatic cells, HPDEs, cannot form a confluent monolayer (data not shown).

Our experiments show that the reduced $S1P_2$ levels in HPAF II cells disrupted apical extrusion, leading to reduced apoptosis rates and enhanced basal extrusion. Similar to HBE monolayers lacking $S1P_2$, HPAF II cells formed masses within a week of culture and displayed extrusion defects and reduced rates of UV-C-induce apoptosis (**Figure 4A–D**). While ~50% of cells did not extrude, most of the remaining cells extruded in the opposite direction—basally, underneath the layer (**Figure 4D,E**) at

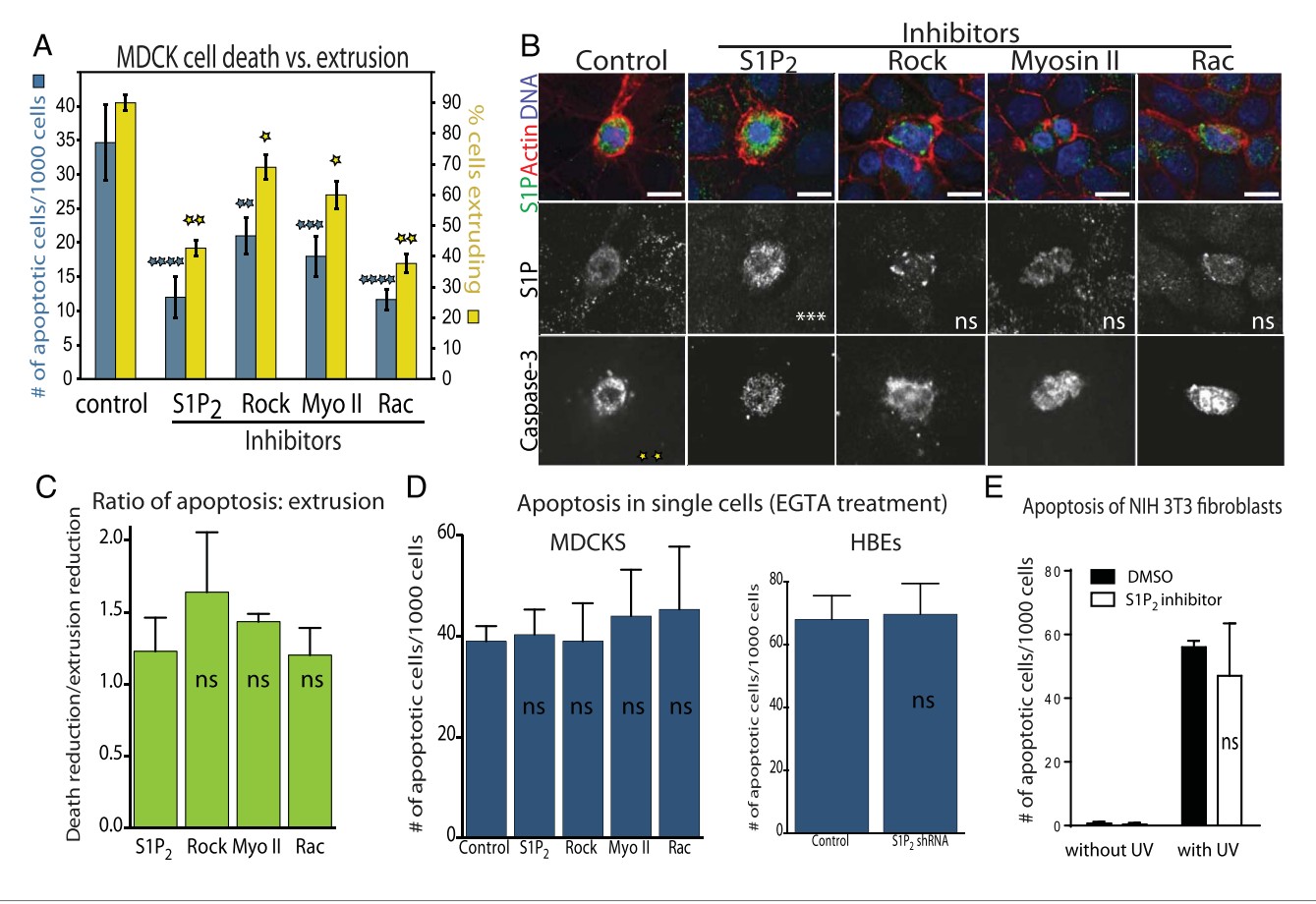

**Figure 3.** Decreased apoptosis is due to blocked extrusion rather than S1P signaling. (**A**) Rates of MDCK cell death (left Y-axis, blue) correspond with cell extrusion rates (right Y-axis, yellow) in response to UV-C when treated with extrusion inhibitors. (**B**) Representative images of apoptotic cells with and without compounds that block extrusion. When extrusion occurs, the dying cell DNA lies above (out of plane from) the neighboring cells with a contracted actin ring but when it fails, it lies in the same plane as surrounding cells with an uncontracted actin ring. Only the S1P$_2$ antagonist JTE013 causes significant S1P accumulation in the dying cell (second column), whereas blocking extrusion with the other compounds does not impact S1P levels, where p values of each drug treatment compared to control are listed on each S1P panel as asterisks (n = 4). Bar = 10 µm. (**C**) Ratio of reduction of extrusion to reduction of apoptosis shows nearly a 1:1 correlation throughout, where p-values compared to S1P$_2$ are not significant. (**D**) Compounds used to block extrusion do not affect apoptosis rates in single MDCK cells treated with EGTA in response to UV. (**E**) Quantification of UV-induced apoptotic NIH 3T3 cells in the presence of vehicle or JTE-013; All results are expressed as mean values ± STD of three separate experiments (*p < 0.01, **p < 0.005, ***p < 0.005, and ****p < 0.0001), and NS in graphs **B**, **D**, and **E** indicate that p values of a unpaired T-test are not significant.

rates similar to when MDCK monolayers are treated with S1P$_2$ antagonist (*Slattum et al., 2014*). Basal extrusion of cells with upregulated survival signaling could potentially enable their invasion beneath the epithelium (*Slattum and Rosenblatt, 2014*). To investigate if basally extruded cells can survive following extrusion (*Slattum and Rosenblatt, 2014*), we analyzed extrusion from three-dimensional cysts, where the fate of basally extruded cells can be followed outside the cyst, rather than beneath a monolayer. MDCKs were used as controls, which, like HPAF II cells, form cysts with hollow apical lumens of 34 µm ± 5 µm in diameter when grown in Matrigel (*Figure 5B*). Approximately 30% of HPAF II cysts extrude cells basally that survive, compared to only ~3% of MDCK cysts (*Figure 5C,D*). Live imaging confirmed that 28.6% of basally extruded cells remained alive throughout a 12-hr video, whereas those from MDCK cysts died during this time (*Video 1*, n = 5 videos of each). Importantly, S1P$_2$-GFP expression is sufficient to rescue apical extrusion (*Figure 5C*), decrease the frequency of live cells that basally extrude (*Figure 5D*), and increase the percentage of cysts with dead cells in their lumens (*Figure 5E*). These data suggest that the S1P-S1P$_2$ signaling required for extrusion is critical not only for promoting cell death but also for preventing basal extrusion, which could enable cells to invade.

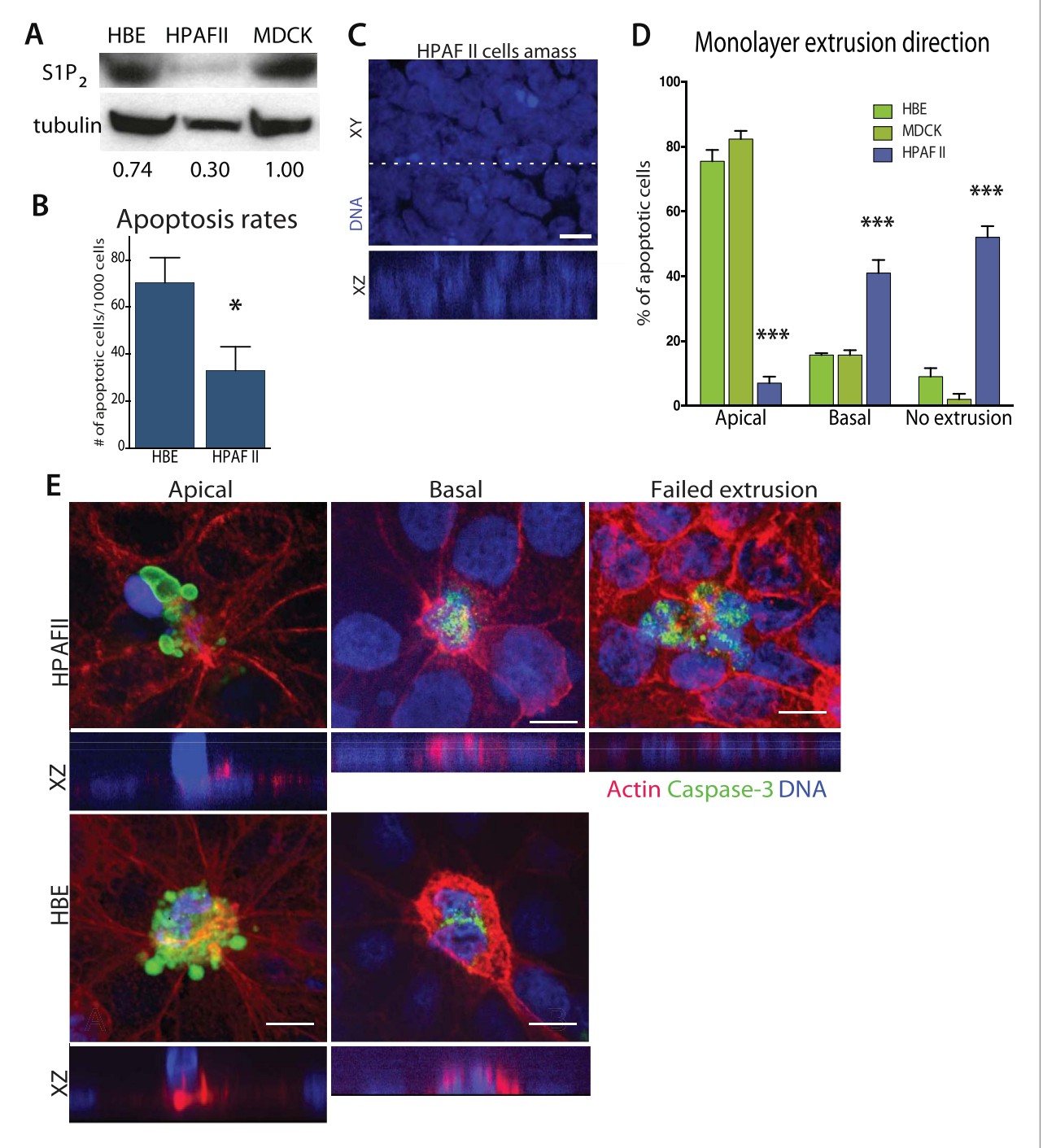

**Figure 4**. Pancreatic cancer cell line HPAF II accumulates into masses and extrudes basally. (**A**) $S1P_2$ immunoblot of HBE (left), HPAF II (middle), and MDCK (left) cells with α-tubulin as loading control. (**B**) Cell death rates in response to UV-C. (**C**) Quantification of cell extrusion events from three independent experiments; n = 300 apoptotic cells per cell line, error bars are STD where *<0.01 and ***<0.0001. (**D**) Representative confocal projection and XZ cross-section (from region in dashed line above) of HPAF II cells that grew into masses rather than monolayers. (**E**) Representative confocal projections of HPAF II (upper panel) and HBE (lower panel) cells undergoing apical (left) or basal (middle) extrusion, with XZ sections below. Basal extrusion was scored when an actin ring contracted above the dying cell (marked by DNA and caspase-3 staining) and the DNA of the dying cell lies in the same plane as the neighboring cells. Scale bars, 10 μm.

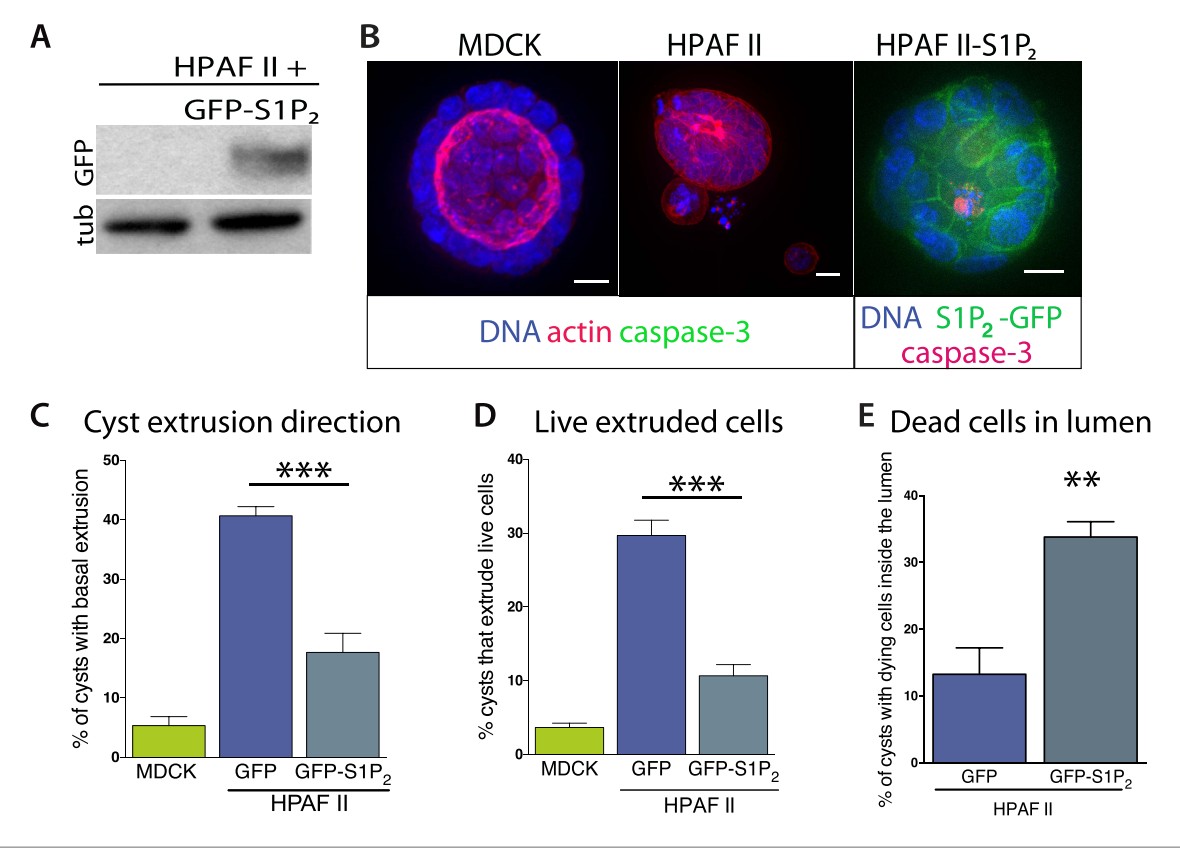

**Figure 5**. Exogenous expression of $S1P_2$ rescues apical extrusion and cell death. (**A**) GFP immunoblot of HPAF II cells expressing $S1P_2$ GFP. (**B**) Representative confocal projections of MDCK, HPAF II, and HPAF II-$S1P_2$ cysts, where scale bar = 10 μm. (**C**) Percentages of MDCK, HPAF II GFP, and HPAF II $S1P_2$ cysts with basal extrusion; n = 300 cysts per cell line. (**D**) Quantification of MDCK, HPAF II GFP, and HPAF II $S1P_2$ cysts extruding live cells basally; n = 300 cysts per cell line. (**E**) Frequency of HPAF II GFP and HPAF II $S1P_2$ cysts with dying cells inside the lumen; n = 300 cysts per cell line. All results are expressed as mean values ± STD of three separate experiments (**p < 0.01, and ***p < 0.001).

## Inhibiting FAK rescues cell death rates in epithelia defective in extrusion

Because extrusion promotes cell death in response to apoptotic stimuli (*Rosenblatt et al., 2001*; *Andrade and Rosenblatt, 2011*) and during normal homeostasis (*Eisenhoffer et al., 2012*; *Marinari et al., 2012*), we hypothesized that it does so by eliminating the competing survival signaling associated with cell-matrix attachment. If so, increased cell survival when extrusion is blocked would derive from prolonged cell attachment to the underlying matrix. Since Focal Adhesion Kinase (FAK) is critical for matrix-dependent survival (*Frisch et al., 1996*), we investigated if FAK were increased in cells targeted for death when extrusion was blocked. Surprisingly, we found that control MDCK cells in early stages of extrusion have far higher levels of active FAK, by immunostaining with a phospho-FAK antibody, than surrounding live cells but that these levels decrease during later stages of extrusion (*Figure 6A*). This increase in pro-survival phospho-FAK in cells targeted to die mimics the increased levels of S1P, another pro-survival signal, in cells triggered to extrude and die that also decrease once cells extrude (*Figure 3C* and [*Gu et al., 2011*]). However, late-staged apoptotic cells (detected by piknotic DNA) still have high levels of active FAK when extrusion is blocked with $S1P_2$ antagonist (*Figure 6A*). This paradoxical survival signaling increase in cells targeted to die may reflect cell-intrinsic compensatory signals to apoptotic signaling that eventually decrease as cells commit to apoptosis. The fact that these survival signals stay high when cell extrusion is blocked suggests that this increased survival signaling derives from inability to detach from matrix.

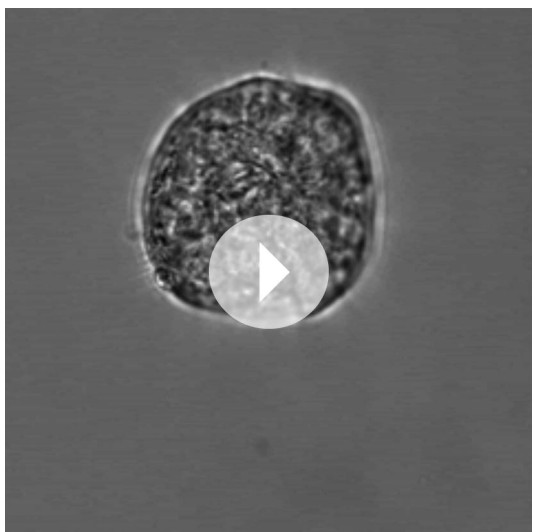

**Video 1.** An HPAF II cyst growing without apoptotic stimuli extrudes live cells basally. Note that some cells extrude and die while others survive and/or migrate away. DOI: 10.7554/eLife.04069.008

Because FAK activity remains high in cells where extrusion is blocked, we wondered if chemically blocking FAK activation would rescue the apoptosis rates of monolayers lacking functional $S1P_2$ signaling. The specific FAK inhibitor PF 573228 had no effect on untreated or UV-treated wild type monolayers (*Figure 6B* and [*Slack-Davis et al., 2007*]), likely because cells not targeted to extrude have quite low levels of active FAK (*Figure 6A*). However, this same FAK inhibitor rescued the cell death rates of monolayers where extrusion is blocked with $S1P_2$ antagonist JTE013 or Rac inhibitor to those seen in wild type monolayers (*Figure 6B*). FAK inhibitor dramatically decreased phospho-FAK staining in cells where JTE013 blocks extrusion (*Figure 6A*). Over-expression of a dominant negative FAK isoform (FRNK) (*Park et al., 2004*) acted similarly to FAK inhibitor (*Figure 6C*). Importantly, FAK inhibitor alone induced apoptosis of HPAF II cells in monolayers (*Figure 6D*) and even those basally extruding from HPAF II cysts (*Figure 6E,F*). Additionally, treatment with FAK inhibitor to cysts that had already accumulated live basally extruded cells was sufficient to nearly double apoptosis rates of basally extruded cells (from 15 ± 2% to 28 ± 3%) when added for only 2 hr.

## FAK inhibitor eliminates $S1P_2$ mutant zebrafish embryo epidermal masses and improves epidermal barrier function

Because inhibition of FAK appears to promote cell death of only extrusion-defective epithelial cells without affecting normal epithelia, we wondered if FAK inhibition could eliminate the epidermal masses of Mil zebrafish embryos as they formed without adversely impacting the animal. While FAK inhibitor had no visible effect on wild-type zebrafish at 5 dpf, it greatly decreased the number and size of epidermal cell masses in $S1P_2$-mutant embryos of the same age (*Figure 7A–C*). While FAK inhibitor treatment sloughed off many of the epidermal cells, which could be found at the bottom of the dish, it also increased the apoptosis rate within the masses (*Figure 7D*). To test if FAK inhibitor could eliminate epidermal masses *after* they form, we needed to inducibly knockdown $S1P_2$ later in development, since Mil zebrafish mutants die due to heart defects around the time cell masses form (~5 dpf; [*Kupperman et al., 2000*]). To do so, we used we photo-activated $S1P_2$ morpholino at 24 hpf (see [*Eisenhoffer et al., 2012*] for characterization of this method) to knockdown $S1P_2$ after heart development occurred. $S1P_2$ morphants had 0.18-fold lower $S1P_2$ protein levels (*Figure 7E*) and phenocopied the epidermal masses seen in Mil mutants (*Figure 7F*). Addition of FAK inhibitor to an $S1P_2$ morphant at 5 dpf with epidermal masses caused the masses to slough off within 19 hr (*Figure 7F*, where n = 6 videos total). Remarkably, we found that while *Mil* embryos had poor epidermal barrier function, as assayed by Texas Red-Dextran$^{MW70}$ permeability, epidermal permeability was significantly reduced with FAK inhibitor (*Figure 7A*). These results suggest that FAK inhibitor alone may selectively target masses resulting from cells defective in extrusion and improve overall epithelial integrity without affecting the normal surrounding tissue.

## Expression of $S1P_2$ is sufficient to reduce orthotopic pancreatic tumors and their metastases

We have shown that disrupting $S1P/S1P_2$ signaling inhibits epithelial cell death, causes masses, and promotes a potential mechanism for invasion—basal extrusion, which together could promote tumor formation and progression. Yet, it was not clear if this signaling pathway plays a role in malignancy. To test the role of $S1P_2$ in tumorigenesis, we orthotopically transplanted HPAF II tumor cells expressing either GFP or $S1P_2$-GFP into nude mice and found that $S1P_2$ expression was sufficient to markedly reduce both tumor size and metastatic frequency (*Figure 8A,B*).

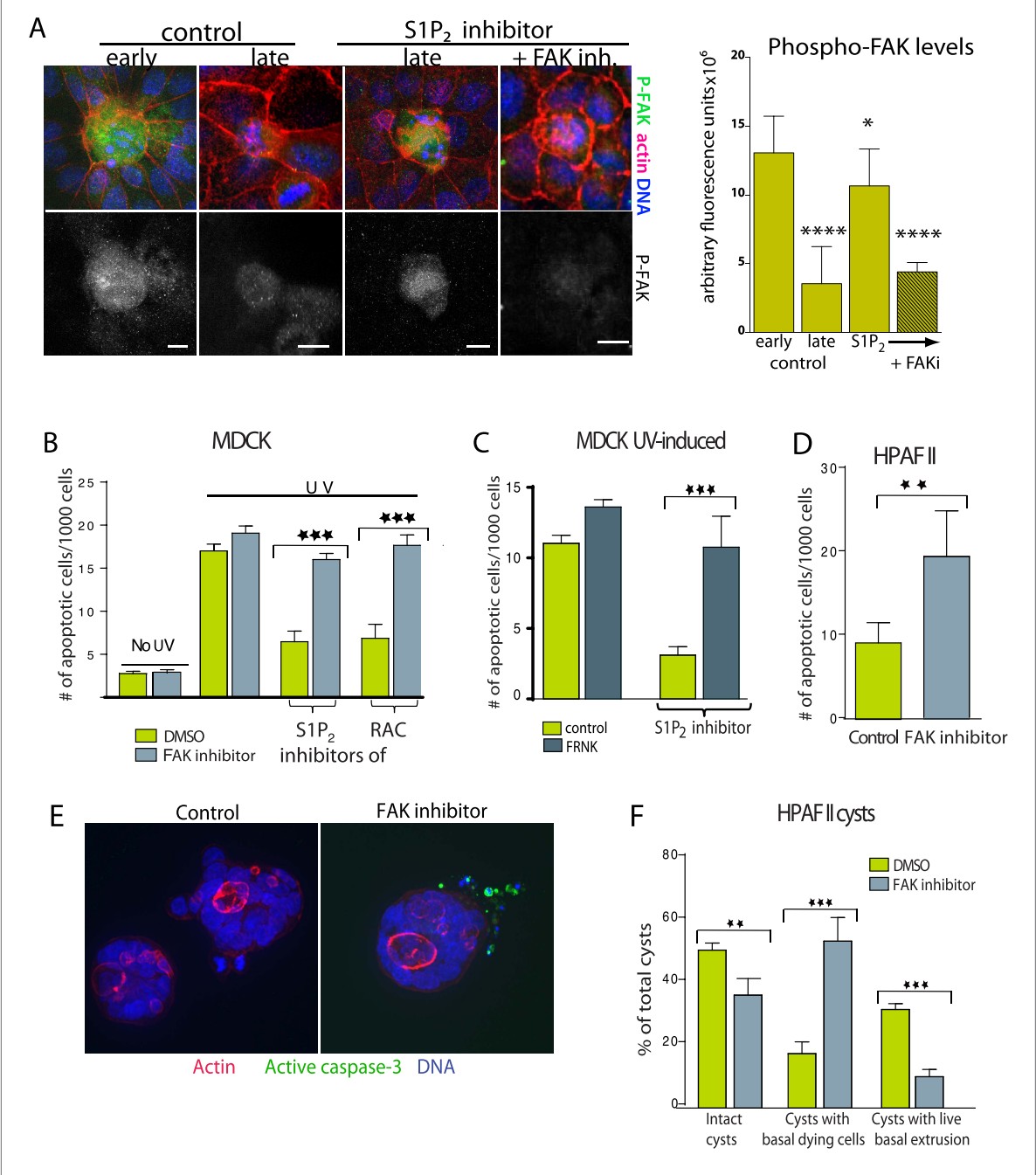

**Figure 6**. Inhibition of FAK activity specifically increases cell death in epithelial cells lacking S1P$_2$. (**A**) Immunostaining of active phospho-FAK in early and late control extrusions and in a JTE-013 (S1P$_2$ antagonist)-inhibited extrusion with late apoptotic cell or one with the FAK inhibitor PF573228, with averaged arbitrary fluorescence units and their p-values compared to early extruding cells in graph on right (n = 10 measurements each over three separate experiments). (**B**) Quantification of UV-induced MDCK apoptosis in the presence of control, JTE-013, or EHT1864 with or without treatment of the FAK inhibitor PF573228, where n = 3000. (**C**) Quantification of UV-induced apoptosis of MDCK cells and those expressing FRNK, where n = 3000. (**D**) Quantification of PF573228-induced apoptosis of HPAF II cells, where n = 3000. (**E**) Representative confocal projections of HPAF II cysts treated with control or PF573228. Scale bars = 10 µm. (**F**) Frequencies of HPAF II cysts with dying cells, live extruding cells, or neither, where n = 300. All quantification results are expressed as mean values ± STD of three separate experiments (*p < 0.05, **p < 0.01, ***p < 0.001, and ****p < 0.0001).

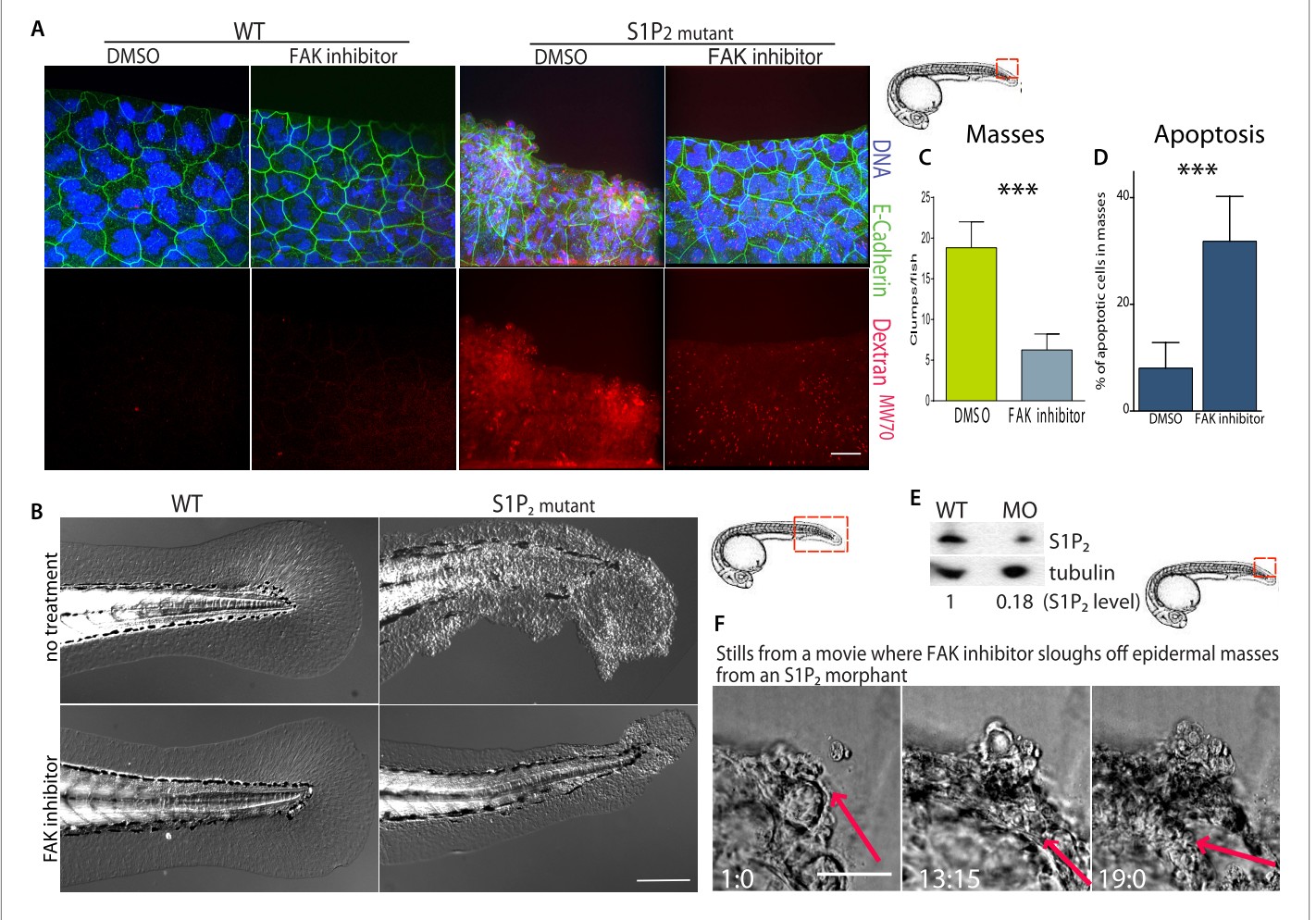

**Figure 7**. FAK inhibitors eliminate epidermal cell masses in $S1P_2$ zebrafish mutants and improve epidermal barrier function without affecting wild type zebrafish. (**A**) Representative confocal projections of 5-dpf WT (left) and *Mil* ($S1P_2$ mutant) (right) zebrafish larvae treated with DMSO or FAK inhibitor PF573228, where high Texas-Red Dextran indicates poor permeability in *Mil* but is greatly reduced when barrier function is improved with FAK inhibitor treatment. Scale bar = 10 µm and red box indicates region of fish imaged. (**B**) 5 dpf *Mil* and wild type zebrafish treated with and without FAK inhibitor. Note that while FAK inhibitor-treated *Mil* have other developmental defects (heart and circulation), there are no obvious clumps as seen in the untreated fish. Scale bar = 100 µm and red box indicates region of fish imaged. Note FAK inhibitor does not affect WT zebrafish. (**C**) Quantification of epidermal masses in 5 dpf *Mil* zebrafish larvae with and without PF573228. (**D**) Quantification of apoptotic cells within epidermal masses with and without PF573228. For both, error bars are SD and p values are ***<0.0001. (**E**) Immunoblot showing knockdown of $S1P_2$ by photo-activatable morpholinos. (**F**) Stills from a video where PF573228 was added to $S1P_2$ morphant at 5 dpf, where red arrows show the edge of the epidermis over time, scale bar = 50 µm and red box indicates region of fish imaged. Time is hours:minutes following FAK inhibitor addition. Note: epidermal cells that are sloughed off become embedded in the agarose where fish is mounted.

## Human pancreatic ductal carcinomas lack S1P₂ receptor

Further, we found that human pancreatic carcinomas have strikingly down-regulated $S1P_2$ protein levels. Because tumors typically have different stromal to epithelial ratios compared to uninvolved pancreatic tissue that can confound microarray data, we immunostained fixed tissue slices for both $S1P_2$ and cytokeratin to highlight epithelial cells so that we could compare $S1P_2$ protein levels in the epithelia alone (***Figure 9A***). We found that $S1P_2$ was significantly lower in pancreatic cancer (PDAC) cells compared to epithelial acini from uninvolved neck margins, from which PDACs may arise, or to pancreatic intraepithelial neoplasia (PanIN) precursor lesions (***Figure 9A–C***). Importantly, lower $S1P_2$ expression correlates with later tumor stages in both averaged (***Figure 9B***) and five patient-matched samples (***Figure 9C***). Loss of $S1P_2$ expression in pancreatic cancers suggests that defective extrusion may contribute to human PDAC development.

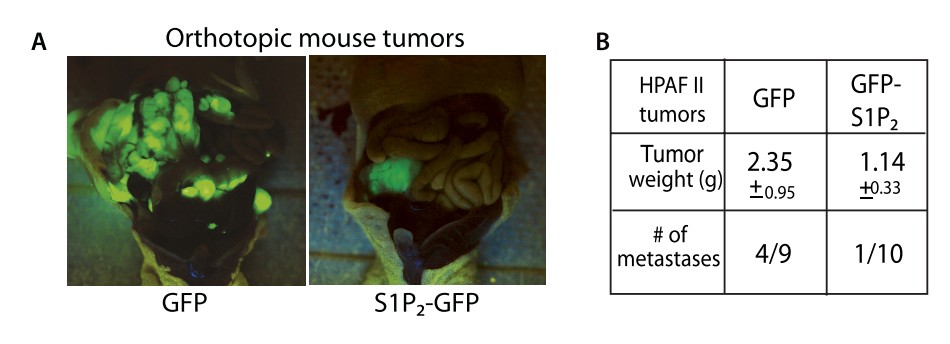

**Figure 8**. Exogenous S1P$_2$ expression reduces orthotopic pancreatic tumors and rates of metastasis in mice. (**A**) Representative images of HPAF II GFP and HPAF II S1P$_2$ orthotopic xenograft tumors in nude mice. (**B**) Summary of tumor weights and metastatic frequency.

## Discussion

Our work presents a new paradigm for how an aggressive class of carcinomas may form and progress: failed extrusion (*Figure 10*). Normal epithelial cells produce S1P to trigger their extrusion and death once they become too crowded, simultaneously maintaining correct cell density and barrier function. Epithelia with defective S1P/S1P$_2$ signaling cannot extrude apically. Extrusion-defective epithelia retain cells, which can result in resistance to homeostatic and chemotherapy-induced cell death and neoplastic masses. Further, a small number of cells also die without getting extruded, which can disrupt barrier function. Aside from allowing access of inappropriate signals, poor epithelia barrier function could cause chronic inflammation—an important factor for tumor progression (*Coussens and Werb, 2002*). Additionally, defective apical extrusion signaling shifts extrusion basally, which could allow transformed cells to invade the underlying tissue (*Slattum and Rosenblatt, 2014*).

Basal extrusion may be a common hallmark of invasive tumor types. We have recently discovered that oncogenic KRas[V12] expression degrades S1P through autophagy and causes cell masses and basal extrusion, similar to the extrusion defects observed when S1P$_2$ is absent (*Slattum et al., 2014*). KRas[V12] is an important driver for the same cancers that lack S1P$_2$—pancreatic, lung, and colon carcinomas— and its expression alone reduces S1P$_2$ (*Slattum et al., 2014*), which may explain why PanIN precursors have reduced S1P$_2$ expression. Further, we have found that another oncogenic mutation, truncation of the adenomatous polyposis coli gene, also results in increased basal extrusion. While it is not clear what mechanisms drive tumor cell invasion, our work showing that exogenous expression of S1P$_2$ can dramatically reduce basal extrusion rates and orthotopic tumor metastasis rates in tumor cells that lack this receptor suggests that S1P$_2$-mediated extrusion may play an important role in metastatic cell invasion.

Because cancer cells lacking S1P$_2$ have increased survival signaling due to an inability to detach from the matrix and its associated survival signaling through increased active FAK, we found that FAK inhibitor on its own could rescue cell death rates to those seen in wild type cells (*Figure 7D*). Surprisingly, FAK inhibitor could also reverse other extrusion defects, such as poor barrier function and survival of basally extruded cells, factors that together could contribute to tumor progression. The fact that adding FAK inhibitor can rescue these defects when added after they form further supports the notion that anoikis results from extrusion and also suggests that FAK inhibitors may be particularly good at treating pancreatic and other carcinomas defective in extrusion. Moreover, we expect a specific FAK inhibitor to not cause the common toxicities associated with standard chemotherapies, as it does not affect normal epithelial tissue. Another FAK inhibitor, PF-00562271, reduces tumor growth, metastases, and ameliorates tumor microenvironment when used in an orthotopic mouse model for pancreatic cancer (*Stokes et al., 2011*), suggesting this drug could be promising for pancreatic cancer patients. However, it is important to note that PF-00562271 also inhibits a FAK-related kinase Pyk-2, which promotes non-specific cell death (*Schultze and Fiedler, 2011*). The phase I clinical trial for PF-00562271 showed that while this drug was tolerated fairly well, it did cause nausea, vomiting, and diarrhea (*Infante et al., 2012*), symptoms indicative of poor gut barrier function, likely due to

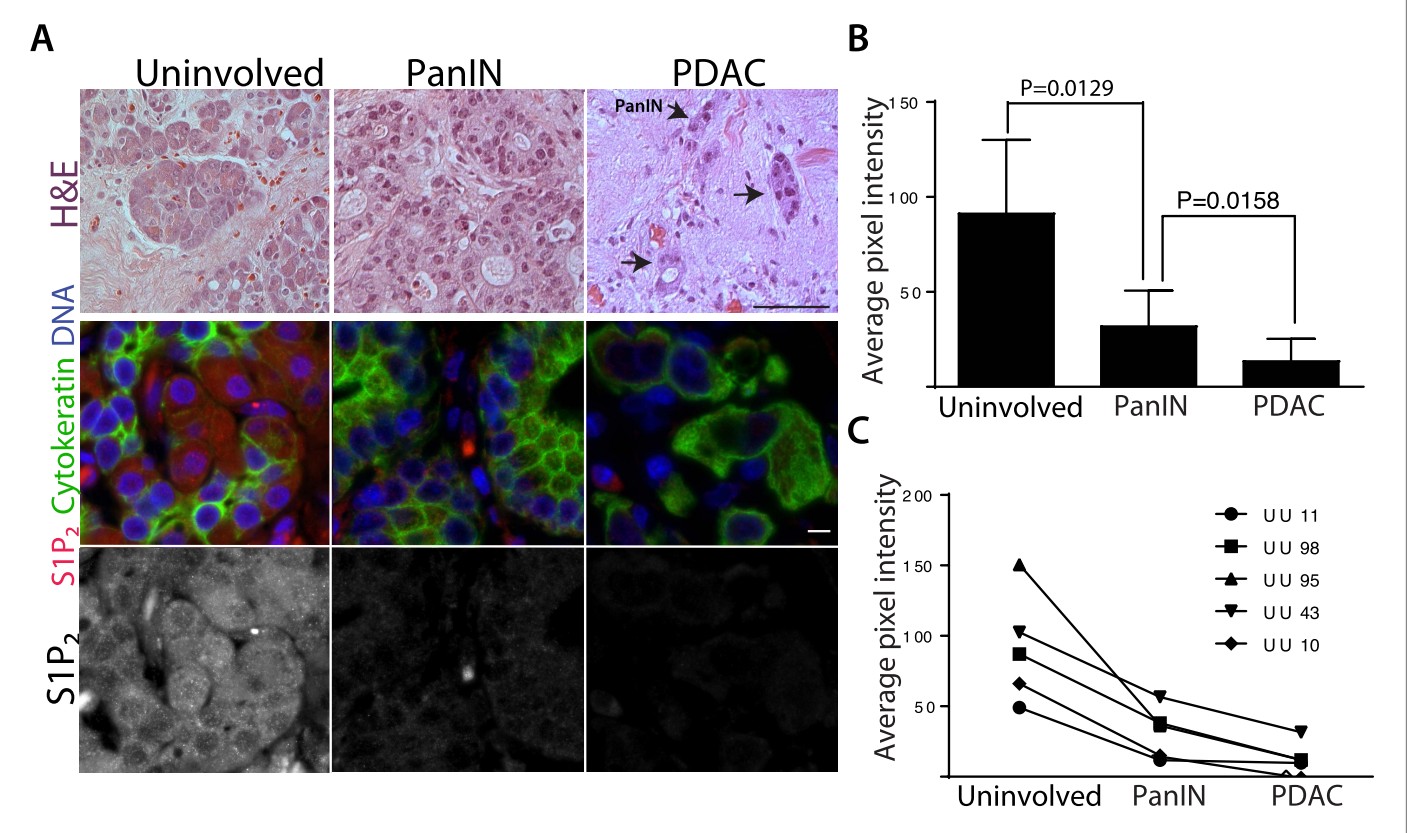

**Figure 9**. Human pancreatic tumors have reduced S1P$_2$ expression. (**A**) H&E (top panel) and confocal fluorescence images (middle and bottom panel) of normal acinar cells from uninvolved neck margin, PanIN, and invasive cancer cells. Scale bars, 100 and 10 μm, respectively. (**B**) Quantification of S1P$_2$ fluorescence intensity in acinar cells, PanIN, and invasive cancer cells from five individual patients. p values were calculated with a paired t test. (**C**) Changes of S1P$_2$ fluorescence intensity from normal acinar cells to invasive cancer cells in each individual patient.

excessive non-specific apoptosis. Our results suggest that a newer more specific FAK inhibitor, such as VS-4718 (*Shapiro et al., 2014*), may provide a more targeted therapy for patients with pancreatic and lung carcinomas that have aberrant extrusion signaling without the common toxicities associated with older chemotherapies.

Based on our previous findings that extrusion drives normal epithelial cell turnover, we have found that disruption of extrusion may contribute to a class of cancers with poor prognosis. We find that cancer cells that lack the extrusion-signaling axis not only have reduced cell death rates but also have poor barrier function and a propensity to extrude cells basally, properties that could lead to higher invasion and metastatic rates. Therefore, aberrant extrusion signaling in pancreatic and lung carcinomas could not only contribute to tumor initiation but also progression. Importantly, specific inhibition of FAK, which does not disrupt normal epithelial tissue, is sufficient to reverse all of the effects of disrupted extrusion and could provide a better, less toxic therapy for this aggressive class of tumors.

## Materials and methods

### Cell culture

MDCK II cells were cultured in Dulbecco's minimum essential medium (DMEM) high glucose with 5% FBS (all from HyClone, Logan, UT) and 100 μg/ml penicillin/streptomycin (Invitrogen, Grand Island, NY) at 5% CO$_2$, 37°C. HBE cells were cultured in MEM supplemented with 10% FBS and l-glutamine in a flask coated with human fibronectin type I (BD, Franklin Lakes, New Jersey), bovine collagen I (Advanced BioMatrix, San Diego, CA), and BSA (Invitrogen). HPAF II cells were cultured in MEM (HyClone) supplemented with 10% FBS.

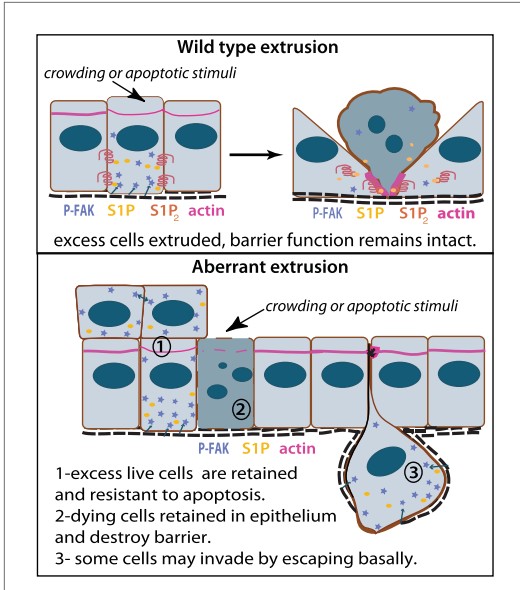

**Figure 10**. Model for how extrusion can promote cell death and suppress tumor formation. Apical extrusion promotes death of grey-blue cell (top panel). Here, pro-survival signals phospho-FAK and S1P (which also promotes extrusion) increase in an early extruding cell but decrease once a cell is extruded and targeted to die (right, cell with piknotic nucleus). However, when apical extrusion is blocked due to lack of $S1P_2$ receptor (bottom panel), epithelial cells do not die and can accumulate (left cell and those accumulating above) from increased matrix-derived survival signaling (arrows from matrix neighboring cells signaling to P-FAK). Additionally, cells can still basally extrude, which could potentially enable their invasion beneath the layer (right cell). Basally extruded cells may also have high P-FAK, since they are sensitive to FAK inhibitor when extruded into matrix in vitro, yet this point will be critical to test in vivo in disseminating tumors. Other cells may still die but not extrude (grey-blue cell with piknotic nucleus), leading to poor barrier function and inflammation, which could also promote tumor progression.

Culturing HPAF II and MDCK cells in Matrigel generated HPAF II and MDCK cysts, respectively. Briefly, a single cell suspension of HPAF II cells was resuspended in Growth Factor Reduced Matrigel (BD Biosciences) final concentration 4% and placed in eight well coverglass chambers (Nalge Nunc, Rochester, NY) coated with a thin polymerized layer of Matrigel. For live imaging, cells were placed on 24 well glass-bottom culture dishes (MatTek Corporation, Ashland, MA). After 20 min incubation at 37°C, cell-growth medium was added on top. Cysts were allowed to grow for the indicated duration and analyzed by time-lapse imaging or fixed with 4% paraformaldehyde for immunostaining.

## Generation of stable Doxycyclin-inducible MDCK II cells over-expressing dominant negative FAK

We first transfected MDCK II cells with neomycin-resistant pTet-ON regulator plasmid, encoding rtTA protein (reverse tTA, tetracycline-controlled transactivator). The stable transfected MDCK II cells were selected by cultivation in media containing 500 µg/ml G418. We then transfected Tet-ON MDCK II cells with FAK-CD-TRE-2-hyg plasmids (*Golubovskaya et al., 2009*) and selected for stably transfected cells with 0.1 mg/ml hygromycin. Expression of FAK-CD was induced with 2 µg/ml doxycycline for 4 days.

## UV and drug treatment

MDCK II, HBE, or HPAF II cells grown to confluence on glass coverslips were exposed to 1200 µJ/cm$^2$ UV$^{254}$ using a Spectrolinker (Spectroline, Westbury, NY) to induce apoptotic extrusion and incubated for 2 hr before fixation. Cells were treated with 10 µM JTE-013 (Tocris Bioscience, United Kingdom), 10 µM Y-27632, 10 µM Blebbistatin, 10 µM EHT 1864 (all from Sigma–Aldrich, Saint Louis, MO), 10 µM PF 573228 (Tocris Bioscience) or 1% DMSO as a control for 10 min before UV treatment. To induce apoptosis with chemotherapy drugs, cells were treated with 20 µg/ml Cisplatin, 1 µM Gemcitabine, or 20 µg/ml 5-Fluorouracil (all from Sigma–Aldrich) for 24 hr.

## Cell staining

Cells were fixed with 4% formaldehyde in PBS at 37°C for 20 min, permeabilized for 10 min with 0.5% Triton in PBS, blocked with AbDil (PBS with 0.1% Triton X-100 and 2% BSA) for 10 min, and incubated with primary antibody for 1 hr. Antibody concentrations used for immunostaining were: 1:200 rabbit anti-active caspase-3 (BD), 1:100 rabbit anti-phospho FAK (Tyr 397) (Cell Signaling, Danvers, Massachusetts), and 50 µg/ml anti-S1P mAb (LPath Inc., San Diego, CA). Alexa Fluor 488 goat anti–rabbit IgG and Alexa Fluor 488 goat anti–mouse IgG were used as secondary antibodies to detect active caspase-3 and S1P, respectively. Actin was detected with Alexa Fluor 568–phalloidin (Invitrogen). DNA was detected with 1 µg/ml Hoechst 33,342 (Sigma–Aldrich).

## Human pancreatic sections

Pancreatic tissue sections (3 µm) were generated from formalin-fixed, paraffin-embedded tissue collected from PDAC patient resections at the University of Utah Hospital. The sections were deparaffinised and rehydrated by incubating in citrus clearing solvent (CCS; Richard Allen Scientific, Kalamazoo, MI), 100%, 95%, 80%, 70% ethanol, and PBS. For immunofluorescence, antigens were retrieved by heating the slides in boiling 10 mM sodium citrate for 20 min, then rinsed three times with PBS, blocked with 5% BSA/0.5% Tween-20 in PBS for 4 hr, and incubated overnight with anti-S1P$_2$ (Imgenex, San Diego, CA) and anti-pan cytokeratin (Sigma–Aldrich) at 4°C, rinsed five times with PBS, incubated in Alexa-488 anti-mouse antibody, 1 µg ml$^{-1}$ Hoechst, and Alexa-568 anti-rabbit antibody for 2 hr, rinsed three times in PBS, and mounted in Prolong Gold (Invitrogen). Fluorescence micrographs of stained slides were obtained using a Leica DM 6000B microscope and captured using a Micromax charge-coupled device camera (Roper Scientific, Sarasota, Florida). IPlab Software was used to control the camera and to process images. Pixel intensity interested area was measured with ImageJ. The University of Utah Institutional Review Board approved the use of human tissue in this study. Tissue sections were obtained from excess clinical pathology tissue from patients, deidentified, resected for pancreatic adenocarcinoma at the University of Utah Huntsman Cancer Institute with appropriate informed consent for use of samples for research purposes (IRB_00010924).

## Zebrafish treatment and staining

We sorted *Mil* zebrafish embryos from their WT homozygous and heterozygous siblings at 2 days post-fertilization (dpf) by the presence or absence of tail blisters. We then treated half of *Mil* mutants and half of the wild type siblings with FAK inhibitor (10 µM PF 573228) for 3 days and added Texas Red-Dextran$^{MW70}$ 30 min before fixation. Embryos were then fixed in PBS with 4% formaldehyde and 0.1% Triton X-100 overnight, blocked with 2 mg/ml BSA for 2 hr, and stained for anti-E Cadherin (Gentex, Zeeland: MI) for 4 hr followed by incubation with Alexa Fluor 488 anti–rabbit IgG Ab. DNA was visualized using 1 µg/ml DAPI.

## Zebrafish morpholino

The antisense morpholino oligonucleotides and photo-morpholino oligonucleotides were acquired from Gene Tools, LLC (Philomath, OR). For the photo-morpholino experiments, the translation blocking antisense morpholino (4 ng/embryo of each) was mixed at a 1:1 molar ratio with a 25 bp sense photo-morpholino and injected into 1–2-cell-stage wild-type AB zebrafish embryos. At 28–32 hpf, embryos were exposed to 350 nm light for 20 s to release the caging sense morpholino, then treated with 10 µM PF 573228 (FAK inhibitor), and filmed by timelapse video microscopy on a spinning disc confocal.

## Image and video acquisition

Confocal Imaging was performed in a Nikon Eclipse TE300 inverted microscope converted for spinning disc confocal microscopy (Andor Technologies, United Kingdom) using a 40× Nikon Apo LWD lenses. Images were acquired with an electron-multiplied cooled charge-coupled device camera (DV887 1004X1002; Andor Technologies) driven by Andor IQ2 imaging software. All images were processed further using Photoshop and Illustrator (Adobe, San Jose, CA), and QuickTime Pro (Apple, San Jose, CA) software.

Live imaging of HPAF II cysts was taken with an OLYMPUS 1X71 inverted microscope using a 20× lens. The images were taken every 5 min for 12 hr. Temperature was controlled by a Weather station connected to the microscope.

For live imaging, FAK inhibitor treated S1P$_2$ morpholino fish were anesthetized with 0.02% Tricaine in E3, mounted in 1% low melt agarose and imaged on a spinning disc confocal at 20×, capturing a z-series every 2 min (*Eisenhoffer and Rosenblatt, in press*) for 3–6 hr.

## Quantification of apoptosis and cell extrusion

To quantify the frequency of apoptosis within a monolayer, we counted the number of active caspase-3 positive cells still in contact with the monolayer per 1000 cells. We excluded round cells with strong caspase-3 staining that were not associated with monolayers, which were likely extruded well before experimental treatments could have impacted them.

To quantify extrusion, extruding cells were manually scored based on the presence of an actin ring compared to where apoptotic cell localized with respect to its neighboring cells. Apoptotic cells above

the plane of the monolayer with strong actin staining around and below them were defined as apically extruded cells. Apoptotic cells remaining in the monolayer and underneath an apical actin ring were considered basally extruded cells. Active caspase-3-positive cells that were not surrounded by a distinguishable actin ring were defined as non-extruded apoptotic cells.

## Immunoblot analysis

Whole-cell extracts were prepared by resuspending cells in NP40 Cell Lysis Buffer (Invitrogen) plus protease inhibitor cocktail and PMSF (Roche, Switzerland). Proteins were resolved by SDS-PAGE using NuPage gels (Invitrogen), and transferred to polyvinylidene difluoride membrane (Thermo, Waltham, MA). Membranes were blocked with 5% dry milk and probed with anti-tubulin 1:1000 (Sigma–Aldrich) and anti-GFP 1:10,000 (Clontech, Mountain View, CA) or anti $S1P_2$ 1:500 (Santa Cruz, Santa Cruz, CA) and identified using horseradish peroxidase conjugated secondary antibodies and enhanced chemiluminescense.

## Molecular cloning

pLL5.0 is a lentiviral expression plasmid containing a U6 promoter to drive expression of the shRNA sequence and a 5′-long terminal repeat to drive the expression of GFP. Designing and cloning of $S1P_2$-specific shRNA were performed as previously described (*Gu et al., 2011*). The full length of human $S1P_2$ was PCR amplified and ligated into the *EcoRI*/*Bam*HI sites of pLL5.0.

## Lentiviral production and transduction

Retroviral production and infections were as described (*Gu et al., 2011*). Infected HPAF II or HBE cells were sorted for GFP by using a BD FACSAria Cell Sorter.

## Surgical orthotopic pancreatic cancer xenograft mouse model

Animals were handled according to protocols approved by the University of Utah Institutional Animal Care and Use Committee. Mice were anesthetized under isoflurane gas; the abdominal skin and muscle were incised just off the midline and directly above the pancreas to allow visualization of the pancreatic lobes; the pancreas was gently retracted and positioned to allow for direct injection of a 100 µl bolus of $1 \times 10^6$ HPAF II cells expressing GFP or $S1P_2$ GFP using a 1 cc syringe with a 30 gauge needle; the pancreas was placed back within the abdominal cavity; and both the muscle and skin layers were closed. 8 weeks later, mice were sacrificed and xenograft tumors were resected and weighed. Metastatic tumors within abdominal wall, liver, and mesentery were also examined and resected.

## Quantifications

P-FAK was quantified using Nikon Elements as the 'ROI Sum Intensity' on ROI statistics using the same ROI size for each projection micrograph measured, subtracting background average fluorescence. Lumen sizes were also measured using Nikon Elements using 'Radius size' using the largest diameter in the 'Annotations and Measurements' analysis package. The statistical analysis was performed using an unpaired or paired *t* test. Values of $p < 0.05$ were considered significant.

## Acknowledgements

We thank Dr George Eisenhoffer, Alexandra Locke, and Richard Dawson for help on zebrafish experiments, and Drs Katie Ullman, Kim Schuske, and Trudy Oliver for helpful comments on our manuscript and LPath for S1P antibody. This work was supported by a National Institute of Health Director's New Innovator Award 1DP2OD002056-01, an R01 R01GM102169 and University of Utah Funding Incentive Seed Grant to JR and P30 CA042014 awarded to The Huntsman Cancer Institute for core facilities.

## Additional information

### Funding

| Funder | Grant reference number | Author |
| --- | --- | --- |
| National Institute of General Medical Sciences | NIH Director's New Innovator Award 1DP2OD002056- | Jody Rosenblatt |
| National Cancer Institute | Cancer Center Support Grant P30 CA042014 | Yapeng Gu, Jody Rosenblatt |

| Funder | Grant reference number | Author |
|---|---|---|
| National Institute of General Medical Sciences | R01GM102169 | Jody Rosenblatt |

The funders had no role in study design, data collection and interpretation, or the decision to submit the work for publication.

## Author contributions

YG, Identified the S1P extrusion pathway and that it was defective in tumors, did nearly all of the experiments, and wrote the paper, Conception and design, Acquisition of data, Analysis and interpretation of data, Drafting or revising the article; JS, Did mouse orthotopic tumor studies, Helped analyze data, Revised manuscript, Acquisition of data, Analysis and interpretation of data, Drafting or revising the article; GS, Contributed data on Rac inhibitor, Made S1P2 morphant fish and filmed them, Revised manuscript, Acquisition of data, Analysis and interpretation of data, Drafting or revising the article, Contributed unpublished essential data or reagents; MAF, Analyzed and obtained pancreatic cancer tissue samples, Helped interpret slides, Commented on manuscript, Analysis and interpretation of data, Drafting or revising the article, Contributed unpublished essential data or reagents; MA, Did experiments on chemotherapies with cell lines, Analyzed data, Had helpful comments on the paper, Acquisition of data, Analysis and interpretation of data, Drafting or revising the article; SJM, Did surgery on pancreatic cancer patients, Consulted on data, Reviewed the paper, Acquisition of data, Analysis and interpretation of data, Drafting or revising the article, Contributed unpublished essential data or reagents; VMG, Provided FAK reagents and consulting, Revised paper, Conception and design, Analysis and interpretation of data, Drafting or revising the article, Contributed unpublished essential data or reagents; JR, Conceived of many experiments, did some fish experiments, helped analyze data, helped write and edit manuscript drafts, Conception and design, Acquisition of data, Analysis and interpretation of data, Drafting or revising the article

## Ethics

Human subjects: The use of human tissue in this study was approved by the University of Utah Institutional Review Board. Tissue sections were obtained from excess clinical pathology tissue from patients resected for pancreatic adenocarcinoma at the University of Utah Huntsman Cancer Institute with appropriate informed consent for use of samples for research purposes (IRB_00010924). Human tissue sample were deidentified and informed consent was obtained from all study participants. The protocol was approved and monitored by the University of Utah Institutional Review Board.

Animal experimentation: This study was performed in strict accordance with the recommendations in the Guide for the Care and Use of Laboratory Animals of the National Institutes of Health. All of the animals were handled according to approved institutional animal care and use committee (IACUC) protocols (#13-06006) of the University of Utah. The protocol was approved by the University of Utah IACUC board.

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
