## [Decision Letter]

Thank you for sending your work entitled “Defective apical extrusion signaling contributes to aggressive tumor hallmarks” for consideration at *eLife*. Your article has been favorably evaluated by Fiona Watt (Senior editor) and three reviewers, one of whom is a member of our Board of Reviewing Editors.

The Reviewing editor and the other reviewers discussed their comments before we reached this decision, and the Reviewing editor has assembled the following comments to help you prepare a revised submission.

This manuscript examines how dysfunction in Sphingosine 1-phosphate receptor (S1PR2) signaling may contribute to cellular overgrowth and tumor biology. All three reviewers found the topic important and the findings presented interesting. The authors demonstrate that disruption of S1PR2 leads to local cellular overgrowth both in cultured mammalian epithelial and zebrafish models. They show that this coincides with loss of both cell extrusion and apoptosis and use a variety of experimental approaches to argue that the primary defect is a failure of extrusion. This is an interesting hypothesis, however, all three reviewers found that the data presented are not sufficient at this point to fully support this conclusion.

The reviewers believe that a revision may be able to address this concern. However, the revision would have to provide significant new evidence. Specifically:

1) The principal reservation is that the data does not fully exclude the possibility that the “overgrowth” is due to a primary effect on the control of apoptosis that is separate from extrusion. Indeed, the data presented in Figures 2 and 3 could also reflect a role for S1P_2_ signaling in apoptotic signaling (a process in which the pathway has been widely implicated) or in damage repair. Thorough control experiments would need to be added to exclude this hypothesis:

A quantification of what percentage of the cell death events reported upon apoptosis-inducing treatments in Figures 2 and 3 results from live extrusion (similar to the quantification on Figure 4) would be required to fully understand the results reported. Indeed, some of the apoptotic stimuli used (e.g. UV) should have an effect independently of the tissue context.

The paper would benefit greatly from experiments to show the time-course of apoptosis following UV damage to determine at which stage the induction of apoptosis is affected.

The authors claim that the experiments with drugs inhibiting actomyosin contractility supports the view that reducing extrusion reduces apoptotic response, as these drugs reduce cell death “to the extent that they inhibit extrusion”. A proper quantification of this effect would be required to support this claim (ratio of death reduction to extrusion reduction?). Particularly because actomyosin contractility has been involved in apoptotic nuclear disintegration (e.g. Croft et al., JCB, 2005) and could thus affect apoptosis independently of extrusion.

Similarly, the observation that blocking S1PR2 doesn't affect apoptosis in 3T3 cells, which don't extrude, does not exclude the possibility that these cells lack the machinery for S1PR2 signaling to regulate apoptosis.

Finally, the authors should test whether S1P2 RNAi and the cytoskeletal inhibitors used affect rates of cell death in the same cells under conditions in which there is no extrusion (e.g. in non-confluent cultures/cultures with no cell-cell adhesion). This will determine whether these pathways alter rates of apoptosis directly in the cell types studied (rather than in NIH3T3).

2) It also remains possible that the overgrowth is due to increased cell proliferation (as well as reduced apoptosis and/or extrusion). Can this be excluded?

3) The HPAFII cyst lumen appears very small compared to MDCK, is this representative and could this explain partly why basal extrusion dominates?

4) Later, the authors show that FAK inhibition eliminates epidermal cell masses. However, how does this fit with the observations from Figure 6, which indicate that inhibition of FAK reduces the survival of basally extruded cells? Does FAK inhibition increase cell death in cell masses? If this is the case, could this be quantified?

5) p-values are missing in several figures (e.g. 3B, 4D, 5D-F).

[Editors' note: further revisions were requested, as described below.]

Thank you for resubmitting your work entitled “Defective apical extrusion signaling contributes to aggressive tumor hallmarks” for further consideration at *eLife*. Your revised article has been favorably evaluated by a member of the Board of Reviewing Editors. The manuscript has been improved but there are some remaining issues that need to be addressed before acceptance, as outlined below:

The authors have addressed many of the points raised by the reviewers. However, one central point remains very unclear: by what mechanism does interfering with S1P_2_ protect from induced (by UV or chemotherapy drugs) apoptosis? This was the main point of the reviewers in the initial review. The authors have now clarified that this protection does not result from reduced live extrusion. The current data and text suggest that the protection results from prolonged attachment of cells and thus exposure to competing survival signaling through FAK. For example, the text states: “Because extrusion normally drives cell death, could it also help promote apoptosis in response to apoptotic stimuli by eliminating competing survival signaling associated with the underlying matrix” (paragraph two in the Results section). However, if this were the case, why does FAK inhibition in control cells not increase cell death upon UV treatment? Would it increase cell death in response to some of the chemotherapy drug treatments? A drawing complementing Figure 10 to explain how they envisage S1P_2_ reduction protects cells against induced apoptosis would greatly clarify the paper.

Without clarification of this mechanistic point, the logics of the argument presented in the paper are confusing and unclear, making it difficult to make a decision concerning the paper. Can the authors clarify this? In other words, is there a link between the role of S1P_2_ signaling in extrusion and the reduction in induced apoptosis when it is not present? If not, the argument presented does not seem to fully make sense and makes the paper look like a series of interesting, but not necessarily related results.

[Editors' note: further revisions were requested prior to acceptance, as described below.]

Thank you for resubmitting your work entitled “Defective apical extrusion signaling contributes to aggressive tumor hallmarks” for further consideration at *eLife*. Your revised article has been favorably evaluated by Fiona Watt (Senior editor) and a member of the Board of Reviewing Editors. We apologise in the delay in getting back to you on this revised version: it took time to get feedback from one reviewer whose expertise is crucial to the paper. The manuscript has been improved but there are a couple of remaining issues that need to be addressed before it can be formally accepted, as outlined below:

The new Figure 6 is very helpful in clarifying how interfering with S1P_2_ might counteract apoptotic signals. While this makes the conclusion of the paper much clearer, these experiments, as displayed, are not fully convincing. How were the levels of phospho-FAK measured? Are the images displayed a confocal slice? If so, in what region of the cell? If not, how do the authors make sure to focus in a comparable region when comparing conditions? We did not manage to find this information in the Materials and methods. The legend indicates that the mean corresponds to n=3. Do they mean the intensities have been quantified in 3 cells per condition? If so, the sample size seems too small to make conclusive statements. Finally, a control showing (hopefully low) P-FAK levels after treatment with the FAK inhibitor would be helpful.

A rather minor point relates to the point on lumen sizes in the previous review round. The authors claim that lumens diameters in MDCK and HPAFII cells are comparable. Where/how was the diameter measured? In the example displayed for HPAFII cells the lumen does not appear to be spherical at all, and while the long diameter is indeed comparable to the MDCK example, the transversal cross section seems to be under 10μm, according to the scale bar. We could not find details on how the diameter was measured in the Methods, could the authors clarify?

---

## [Author Response]

*1) The principal reservation is that the data does not fully exclude the possibility that the “overgrowth” is due to a primary effect on the control of apoptosis that is separate from extrusion. Indeed, the data presented in*
Figures 2 and 3
*could also reflect a role for S1P*_*2*_
*signaling in apoptotic signaling (a process in which the pathway has been widely implicated) or in damage repair*. *Thorough control experiments would need to be added to exclude this hypothesis:*

*A quantification of what percentage of the cell death events reported upon apoptosis-inducing treatments in*
Figures 2 and 3
*results from live extrusion (similar to the quantification on*
Figure 4*) would be required to fully understand the results reported. Indeed, some of the apoptotic stimuli used (e.g. UV) should have an effect independently of the tissue context*.

*The paper would benefit greatly from experiments to show the time-course of apoptosis following UV damage to determine at which stage the induction of apoptosis is affected*.

With respect to these questions, we wanted to clarify that while live cell extrusion can be triggered by crowding (during normal turnover), when epithelia are treated with apoptotic stimuli, they activate apoptotic cell extrusion at much higher rates than the intrinsic homeostatic live cell extrusion. In both cases, S1P signaling through the S1P_2_ receptor drives extrusion. In the first case, cells are extruded alive that later die via Piezo-1. In the second case, cells are extruded as part of the death pathway (likely via caspase activation)—here we find that extrusion also helps promote the death but because apoptosis has been activated, the cells are often dying while they extrude. I think we may need to make this clearer in our article that in Figure 1, cells amass from reduced rates of intrinsic homeostatic extrusion-dependent cell death, whereas in Figures 2 and 3, we induced cell death with a strong apoptotic stimulus, UV-C. When we induce apoptosis, the low intrinsic rates of live cell extrusion become dwarfed by those induced to die—therefore, most cells that do extrude will be caspase-positive. While some cells die when the extrusion pathway is disrupted, it is greatly reduced due to competing cell survival coming from prolonged attachment of cells (that should have died) to the matrix and its survival signaling through FAK.

We see how this point could be confusing and to avoid the confusion, we have changed this sentence of the Results section to the following: *“*We next wondered if extrusion-deficient cells were also more resistant to cell death in response to apoptotic stimuli. […] Because extrusion normally drives cell death, could it also help promote apoptosis in response to apoptotic stimuli by eliminating competing survival signaling associated with the underlying matrix? We find that disrupting extrusion signaling also disrupted apoptosis in response to a variety of apoptotic stimuli*.*”

*The authors claim that the experiments with drugs inhibiting actomyosin contractility supports the view that reducing extrusion reduces apoptotic response, as these drugs reduce cell death “to the extent that they inhibit extrusion”. A proper quantification of this effect would be required to support this claim (ratio of death reduction to extrusion reduction?). Particularly because actomyosin contractility has been involved in apoptotic nuclear disintegration (e.g. Croft et al.,*
*JCB, 2005**) and could thus affect apoptosis independently of extrusion*.

As suggested by the reviewers, we have calculated the ratio of death reduction to extrusion reduction (Figure 3). The results show that the ratios of death reduction to extrusion reduction caused by inhibitors of S1P_2_, Rock, Myosin II, and Rac are all approximately 1:1, supporting the conclusion that these drugs reduce cell death to the extent that they inhibit extrusion.

We agree with the reviewers that actomyosin contractility has been involved in apoptotic nuclear disintegration (Croft et al., JCB, 2005). However, disruption of myosin II contractility with Y-27632 or blebbistatin has no effect on activation of caspase 3 in single cells, which we have added in as Figure 3 by treating similarly-aged MDCK monolayers with these inhibitors in the presence of EGTA to disrupt cell-cell junctions.

*Similarly, the observation that blocking S1PR2 doesn't affect apoptosis in 3T3 cells, which don't extrude, does not exclude the possibility that these cells lack the machinery for S1PR2 signaling to regulate apoptosis*.

Expression of S1P_2_ in NIH 3T3 cell has been shown in many previous publications (5; 18) and we added these references to our paper for clarification.

*Finally, the authors should test whether S1P2 RNAi and the cytoskeletal inhibitors used affect rates of cell death in the same cells under conditions in which there is no extrusion (e.g. in non-confluent cultures/cultures with no cell-cell adhesion). This will determine whether these pathways alter rates of apoptosis directly in the cell types studied (rather than in NIH3T3)*.

Because under-confluent HBE and MDCK cells rarely undergo apoptosis in response to strong apoptotic stimuli, we have instead used EGTA to disrupt cell junction and thus extrusion in similarly aged and confluent cultures. As shown in Figure 3, S1P_2_ shRNA or cytoskeleton inhibitors do not reduce apoptosis rates in HBE and MDCK cells, respectively, suggesting that the effect of inhibiting S1P_2_ or the cytoskeleton on apoptosis depends on extrusion.

*2) It also remains possible that the overgrowth is due to increased cell proliferation (as well as reduced apoptosis and/or extrusion)*. *Can this be excluded?*

We used Biovison Quick Cell Proliferation Colorimetric Assay to compare the proliferation rate between HBE cells expressing control shRNA or S1P_2_ shRNA. We found no difference in the proliferation rates between these two cell lines and have added this graph in Figure 1.

*3) The HPAFII cyst lumen appears very small compared to MDCK*, *is this representative and could this explain partly why basal extrusion dominates?*

HPAF cells form comparable sizes of cysts and lumens to the MDCK cells. In addition, some HPAF cysts could grow bigger than normal size after long-term culture (beyond our images here). The HPAF II cyst shown in Figure 5 looks smaller than the MDCK cyst because a different scale bar is used, as noted in the figure legend, to show live extruded cells surrounding it.

*4) Later, the authors show that FAK inhibition eliminates epidermal cell masses*. *However, how does this fit with the observations from*
Figure 6*, which indicate that inhibition of FAK reduces the survival of basally extruded cells? Does FAK inhibition increase cell death in cell masses? If this is the case, could this be quantified?*

In both cases, we believe FAK inhibition increases the cell death specifically in cells that did not apically extrude. In the case of the HPAF II cysts, the cells that die are embedded in the matrix and have nowhere else to go, so we could quantify them (Figure 6). However, in the zebrafish, we find that cells from the cells comprising the masses in S1P_2_ mutants slough off. While this makes it more difficult to later quantify, we have now added a figure showing stills from a movie demonstrating that the outer clumps detach and become trapped in the agarose after treatment with FAK inhibitor. FAK inhibition also increases cell death in cell masses and we have now added quantification of this as a graph in Figure 7. This is also quantified in Figure 6 in the cysts where FAK inhibitor increases cell death.

*5) p-values are missing in several figures (e.g. 3B, 4D, 5D-F)*.

We have added the appropriate p-values. Thanks for pointing these out.

[Editors' note: further revisions were requested, as described below.]

*The authors have addressed many of the points raised by the reviewers. However, one central point remains very unclear: by what mechanism does interfering with S1P*_*2*_
*protect from induced (by UV or chemotherapy drugs) apoptosis? This was the main point of the reviewers in the initial review. The authors have now clarified that this protection does not result from reduced live extrusion. The current data and text suggest that the protection results from prolonged attachment of cells and thus exposure to competing survival signaling through FAK. For example, the text states: “Because extrusion normally drives cell death, could it also help promote apoptosis in response to apoptotic stimuli by eliminating competing survival signaling associated with the underlying matrix” (paragraph two in the Results section). However, if this were the case, why does FAK inhibition in control cells not increase cell death upon UV treatment? Would it increase cell death in response to some of the chemotherapy drug treatments? A drawing complementing*
Figure 10
*to explain how they envisage S1P*_*2*_
*reduction protects cells against induced apoptosis would greatly clarify the paper*.

*Without clarification of this mechanistic point, the logics of the argument presented in the paper are confusing and unclear, making it difficult to make a decision concerning the paper. Can the authors clarify this? In other words, is there a link between the role of S1P*_*2*_
*signaling in extrusion and the reduction in induced apoptosis when it is not present? If not, the argument presented does not seem to fully make sense and makes the paper look like a series of interesting, but not necessarily related results*.

Apologies, we did not get that this was what you were asking in the first round of reviews and thought that the confusion was instead about extrusion from apoptotic stimuli versus crowding.

We are now adding data that we had previously considered including but didn’t because we were worried that it would confuse readers: when cells are triggered to extrude and die, they, surprisingly, dramatically increase pro-survival active-FAK, as revealed by phospho-FAK immunostaining (Figure 6). While phospho-FAK is high in early-staged extruding cells, it decreases dramatically late in extrusion, as the cells become apoptotic. The same is true for S1P lipid, which is a survival factor in addition to triggering extrusion (Figure 3 and [12]). When extrusion is blocked and cells remain in contact with matrix, phospho-FAK (and S1P) remains high, which is likely why these cells alone are susceptible to FAK inhibitor. While there is a low basal level of phospho-FAK (and S1P) in the surrounding live cells, it seems that cells targeted to die respond by upregulating their survival signaling to compete with the apoptotic signaling. Presumably, apoptosis occurs if the apoptotic signaling wins over survival signaling. We have seen this increase of survival signaling in cells targeted to die routinely, but as it goes against most people’s ideas about apoptosis, were afraid to put it in our manuscript.

We have now included pictures with quantification of P-FAK staining and included this in our schematic. We have quantified the P-FAK immunostaining compared to control surrounding non-extruding cells, with P-values compared to live surrounding cells. We believe that this data should clarify why cells targeted to extrude with upregulated survival signaling (P-FAK) do not die if extrusion is blocked since active FAK remains high from prolonged attachment to the matrix. This also explains why FAK inhibitor exclusively targets cells for death that were blocked from extruding whereas it does not affect neighboring cells with low levels of FAK. We expect to see this same P-FAK staining in response to chemotherapy or crowding-induced extrusion since FAK inhibitor promotes apoptosis to the expected levels as controls when extrusion is blocked during homeostasis.

We have added in the Results section text for Figure 6: “Since Focal Adhesion Kinase (FAK) is critical for matrix-dependent survival (10), we investigated if FAK were increased in cells targeted for death when extrusion was blocked. Surprisingly, we found that control MDCK cells in early stages of extrusion have far higher levels of active FAK, by immunostaining with a phospho-FAK antibody, than surrounding live cells but that these levels decrease during later stages of extrusion (Figure 6). […] The specific FAK inhibitor PF 573228 had no effect on untreated or UV-treated wild type monolayers (Figure 6 (23)), likely because cells not targeted to extrude have quite low levels of active FAK (Figure 6).”

Also, for the schematic (Figure 10), we have changed the legend to say: “Figure 10. Model for how extrusion can promote cell death and suppress tumor formation. Apical extrusion promotes death of grey-blue cell (top panel). Here, pro-survival signals phospho-FAK and S1P (which also promotes extrusion) increase in an early extruding cell but decrease once a cell is extruded and targeted to die (right, cell with piknotic nucleus. […] Other cells may still die but not extrude (grey-blue cell with piknotic nucleus), leading to poor barrier function and inflammation, which could also promote tumor progression.”

[Editors' note: further revisions were requested prior to acceptance, as described below.]

*The new*
Figure 6
*is very helpful in clarifying how interfering with S1P*_*2*_
*might counteract apoptotic signals. While this makes the conclusion of the paper much clearer, these experiments, as displayed, are not fully convincing*. *How were the levels of phospho-FAK measured?*

We used Nikon Elements ROI statistics and measured the Sum Intensity of ROI through each projection. We used the same size ROI for each measurement for consistency and used this number subtracting the average background levels. We repeated the experiment an additional two times (we put n=3, but expect it is 5 but can’t find the old slides) and used 10 images each.

*Are the images displayed a confocal slice? If so, in what region of the cell? If not*, *how do the authors make sure to focus in a comparable region when comparing conditions?*

These are projections from a confocal z series. Further, when we did look at ROI statistics through each section, we saw little variance, suggesting that levels were fairly consistent throughout the cell.

*We did not manage to find this information in the Materials and methods*.

We have added this information in the ‘Quantifications’ section at the end of the Methods section.

*The legend indicates that the mean corresponds to n=3. Do they mean the intensities have been quantified in 3 cells per condition? If so, the sample size seems too small to make conclusive statements. Finally, a control showing (hopefully low) P-FAK levels after treatment with the FAK inhibitor would be helpful*.

We repeated this experiment to add the FAK inhibitor and, indeed, the levels do go down. We had done this experiment 3 times but did not have enough old micrographs to get a higher n for quantifying. So, we have taken more pictures and increased the n that we quantified. This did not change the results at all—we still see that PFAK is high in early control and late blocked extrusions but greatly diminished in late control extrusions and FAK inhibitor-treated dying cells. To increase clarity, we now include a graph of P-FAK quantification next to the pictures in Figure 6.

*A rather minor point relates to the point on lumen sizes in the previous review round. The authors claim that lumens diameters in MDCK and HPAFII cells are comparable. Where/how was the diameter measured? In the example displayed for HPAFII cells the lumen does not appear to be spherical at all, and while the long diameter is indeed comparable to the MDCK example, the transversal cross section seems to be under 10μm, according to the scale bar*. *We could not find details on how the diameter was measured in the Methods, could the authors clarify?*

Measurements here were also done using Nikon Elements ‘Radius’ measurement in the ‘Annotations and Measurements’ package. We have added this to the ‘Quantification’ section of the Methods.

We agree that the cyst in Figure 5 in particular is a bit skewed. I tried to go back and find another one with a more spherical lumen that also shows the cells outside but have had problems with our set up. These pictures have to be taken on a confocal but to find the cyst to scan, we must use the wide-field fluorescence lamp on our confocal, which has been out for a few months. Searching while scanning on the confocal is nearly impossible, so this has made it hard to find another cyst with a perfect lumen that shows the main point that we want to convey-that HPAF II cells extrude basally, divide, and migrate. It’s not easy to capture all of these events in one picture, but feel it important to show the best example of this (the present one), as most people don’t view the movies. While some of the lumens were a bit skewed in both MDCKs and HPAFII, quantification was done on the majority of cysts that were spherical on our wide-field microscope so we could get high numbers. There really is no difference in size.